# Investigations of the Deuterium Permeability of As-Deposited and Oxidized Ti_2_AlN Coatings

**DOI:** 10.3390/ma13092085

**Published:** 2020-05-01

**Authors:** Lukas Gröner, Lukas Mengis, Mathias Galetz, Lutz Kirste, Philipp Daum, Marco Wirth, Frank Meyer, Alexander Fromm, Bernhard Blug, Frank Burmeister

**Affiliations:** 1Department of Tribology, Fraunhofer-Institut für Werkstoffmechanik IWM, Woehlerstrasse 11, 79108 Freiburg, Germany; philipp.daum@iwm.fraunhofer.de (P.D.); marco.wirth@iwm.fraunhofer.de (M.W.); frank.meyer@iwm.fraunhofer.de (F.M.); alexander.fromm@iwm.fraunhofer.de (A.F.); bernhard.blug@iwm.fraunhofer.de (B.B.); frank.burmeister@iwm.fraunhofer.de (F.B.); 2Department of High Temperature Materials, DECHEMA-Forschungsinstitut, Theodor-Heuss-Allee 25, 60486 Frankfurt am Main, Germany; lukas.mengis@dechema.de (L.M.); mathias.galetz@dechema.de (M.G.); 3Department of Epitaxy, Fraunhofer-Institut für Angewandte Festkörperphysik IAF, Tullastraße 72, 79108 Freiburg, Germany; lutz.kirste@iaf.fraunhofer.de

**Keywords:** MAX phase, Ti_2_AlN, PVD coating, oxidation, hydrogen permeation

## Abstract

Aluminum containing M_n+1_AX_n_ (MAX) phase materials have attracted increasing attention due to their corrosion resistance, a pronounced self-healing effect and promising diffusion barrier properties for hydrogen. We synthesized Ti_2_AlN coatings on ferritic steel substrates by physical vapor deposition of alternating Ti- and AlN-layers followed by thermal annealing. The microstructure developed a {0001}-texture with platelet-like shaped grains. To investigate the oxidation behavior, the samples were exposed to a temperature of 700 °C in a muffle furnace. Raman spectroscopy and X-ray photoelectron spectroscopy (XPS) depth profiles revealed the formation of oxide scales, which consisted mainly of dense and stable α-Al_2_O_3_. The oxide layer thickness increased with a time dependency of ~t^1/4^. Electron probe micro analysis (EPMA) scans revealed a diffusion of Al from the coating into the substrate. Steel membranes with as-deposited Ti_2_AlN and partially oxidized Ti_2_AlN coatings were used for permeation tests. The permeation of deuterium from the gas phase was measured in an ultra-high vacuum (UHV) permeation cell by mass spectrometry at temperatures of 30–400 °C. We obtained a permeation reduction factor (PRF) of 45 for a pure Ti_2_AlN coating and a PRF of ~3700 for the oxidized sample. Thus, protective coatings, which prevent hydrogen-induced corrosion, can be achieved by the proper design of Ti_2_AlN coatings with suitable oxide scale thicknesses.

## 1. Introduction

The increasing number of applications in which hydrogen is being used as a storage medium in energy conversion technologies demands the consideration of new construction materials, or at least a profound surface conditioning of established materials to prevent, e.g., hydrogen diffusion induced embrittlement or other forms of corrosion, especially the development of so-called “white etching cracks” [1]. One route for corrosion protection is the development and application of temperature-resistant coatings with excellent barrier properties for hydrogen. Recently performed studies indicate that MAX phase materials might fulfill these requirements [2,3,4,5,6]. The general formula, M_n+1_AX_n_, (short MAX) describes a family of materials consisting of an early transition metal (M), mostly a group 13 or 14 element (A) and nitrogen and/or carbon (X) with the stoichiometry of n = 1,2,3 [7]. The MAX phases crystallize in a hexagonal lattice within the space group D46h (P63/mmc) in which the octahedral M_n+1_X_n_ layers are separated by atomic monolayers of pure A-atoms. MAX phase materials are known to have a good oxidation resistance [3,8,9], a high damage tolerance as well as a high electrical and thermal conductivity [10]. 

The good oxidation resistance of Al containing MAX phases usually stems from the formation of dense and thermodynamically stable thermal grown oxides (TGO) consisting of α-Al_2_O_3_ on the coatings surface at relatively low temperatures of 600–700 °C. For comparison, the direct physical vapor deposition (PVD) of an α-Al_2_O_3_ in an industrial scale deposition process usually requires temperatures above 1000 °C [11]. Lower deposition temperatures of 500–600 °C have also been observed but at the expense of a brittle fracture behavior [12,13]. A further advantage of α-Al_2_O_3_ oxide scales thermally grown on MAX phase coatings is the well-known self-healing effect whereby small defects or cracks in the coating, which might serve as diffusion pathways, are blocked by oxide growth [2]. For this purpose, the oxidation kinetics of the TGO has to allow for a quick healing and oxidation of the surface, but has to prevent fast oxygen diffusion to the coating–substrate interface. The oxidation kinetics of Ti_2_AlC at 1200 °C were modelled by G.M. Song et al. in [3]. This model contains the growth of oxide grains and assumes that the diffusion paths along the grain boundaries increase with time. This results in a time dependency for the increase in the thickness of the oxide scale *d_Ox_(t)* of:(1)$$d_{Ox}\left(t\right)=2\sqrt{k_{n}}\times t^{1/4}$$

The growth factor $k_{n}=\Omega D_{GB}\Delta C\frac{3\delta }{d_{0}}$ contains a constant prefactor *Ω*, the diffusion coefficient for oxygen along the grain boundaries *D_GB_*, the size of the grain boundaries *δ*, the initial lateral grain size *d_0_* and the gradient in the oxygen concentration ΔC.

This model of α-Al_2_O_3_-formation, as well as the structural properties of MAX phases, i.e., the sequence of dense MX-layers, motivated the present investigation on their barrier properties against hydrogen diffusion. 

Although little information about the diffusion of hydrogen in MAX phases exists, similarly composed carbides or nitrides of early transition metals are already used as diffusion barriers for hydrogen [4,14]. It is expected that the anisotropic structure of MAX phases will induce a directional anisotropy of the hydrogen diffusion. In [5], F. Colonna and C. Elsässer presented the findings of an atomistic simulation of diffusion processes in Ti_2_AlN using density-functional theory (DFT). Therein, interstitial diffusion paths of hydrogen and oxygen were examined. It was found that, for hydrogen, the migration perpendicular to the basal planes has a maximum barrier of ~3 eV, whereas the migration barrier parallel to the basal plane is one order of magnitude lower. The high migration barrier parallel to the c axis was explained by the presence of the Ti_2_N double layer, where the interstitial octahedral sites of Al_3_Ti_3_ are already occupied by nitrogen atoms. 

An experimental study on the hydrogen barrier properties of MAX phase coatings was presented by C. Tang et al. in [6]. Therein, ZrY-4 alloy cylinders were coated with Ti_2_AlC and Cr_2_AlC by a multilayer deposition followed by a subsequent annealing step. This process led to a {0001}-textured polycrystalline growth which could also be detected in [15] for Ti_2_AlN. After loading the specimens in an Ar+H_2_ atmosphere the cylinders were investigated by neutron radiography. It could be shown that a 5 µm thick Ti_2_AlC and Cr_2_AlC reduced the penetration of hydrogen under the detection limit.

To evaluate coatings in terms of their capability to reduce the hydrogen permeation a permeation reduction factor (PRF) can be calculated using the mass specific ion current j:(2)$$\mathrm{PRF}=\frac{j_{uncoated}}{j_{coated}}.$$

D. Levchuk et al. investigated Al-Cr-O coatings [16] and Er_2_O_3_ coatings [17] as hydrogen permeation barrier. Both coatings tend to form a dense crystalline structure, which is capable of effectively reducing the hydrogen permeation up to a PRF(Al-Cr-O) = 3500 and PRF(Er_2_O_3_) = 800.

## 2. Experimental Details

### 2.1. Deposition of Ti_2_AlN

The Ti_2_AlN MAX coatings were deposited on AISI 430 ferritic stainless-steel substrates (Fe81/Cr17/Mn/Si/C/S/P), which were polished (1400 grit) and cleaned in acetone and isopropanol using an ultrasonic bath prior to deposition. A custom build industrial sized magnetron sputter chamber SV400/S3 (FHR Anlagenbau GmbH, Ottendorf-Okrilla, Germany) equipped with rectangular titanium (purity 99.8%) and aluminum targets (99.999%) was utilized. To obtain a pronounced {0001}-texture with a parallel orientation of the basal planes and the substrate surface, we alternately deposited 150 single layers of Ti and AlN on the substrate, beginning with Ti. During the radio frequency-sputtering of the aluminum target, nitrogen (purity 99.9999%) was introduced in the chamber. A final subsequent annealing at 700 °C for 1 h in vacuum led to the formation of textured Ti_2_AlN MAX phase coatings. Details of the deposition process are described elsewhere [15]. The coatings thickness was in the 2 µm–3 µm range.

### 2.2. Oxidation Procedure and Analysis

To investigate the oxidation kinetics of the Ti_2_AlN coatings, comparable samples originating from the same batch were oxidized at 700 °C for 5 h, 10 h, 20 h and 100 h in a muffle furnace (Nabertherm GmbH, Lilienthal, Germany) in air. The samples were afterwards removed from the furnace and cooled in air. The crystallographic orientation and phase composition of oxidized and non-oxidized coatings were investigated by X-ray diffractometry (XRD) using a PANalytical Empyrean in parallel beam geometry (Empyrean, PANalytical, Almelo, The Netherlands) and Cu Kα_1_ radiation with a 2-bounce Ge 220 monochromator. The samples were irradiated with primary X-rays using a line focus. The diffracted X-rays were detected using a PIXel-3D detector with a 1 mm slit for the phase analysis. 

A surface sensitive phase analysis was performed using a confocal Raman spectrometer (Model inVia, Renishaw plc., Gloucestershire, United Kingdom) in backscatter geometry. The excitation wavelength λ_Nd:YAG_ = 532 nm was used to determine possible changes in the Ti_2_AlN phase upon thermal treatment whereas the wavelength λ_HeNe_ = 633 nm proofed to be suitable for exciting fluorescence bands in the thermally grown AlO_x_ phase. In all measurements, a 100-fold objective focused the laser on the surface to a spot diameter of about 2 µm.

XPS depth profiles were recorded with a PHI 5000 VersaProbe II (Ulvac-PHI, Inc., Chigasaki, Japan) equipped with an argon sputter option using Al Kα-rays. For analyzing the coarse elemental distribution close to the interface of coating and substrate a metallographic cross-section was prepared after an electrochemical deposition of Ni for protective purposes. The electron probe micro analysis (EPMA) was performed utilizing a JXA-8100 (Jeol, Akishima, Japan). The measurements were performed with an acceleration voltage of 15 kV and a dwell time of 30 ms.

### 2.3. Deuterium Permeation Setup

To investigate the diffusion of deuterium from the gas phase through coated and oxidized membranes, a permeation setup was developed following the works of C. Frank et al. [18], J. Gorman et al. [19] and D. Levchuk et al. [20]. The test rig consisted of two chambers separated by a thin steel membrane (see Figure 1). The high pressure side is filled with the diffusional species or the purging gas, the low pressure side is evacuated by a turbomolecular pump and an ion getter pump down to pressures of ~10^-6^ Pa. The latter is also equipped with a quadrupole mass spectrometer (Model PrismaPlus^TM^ QMG 220, Pfeiffer Vacuum Technology AG, Aßlar, Germany) to determine the gas composition as well as the mass and time dependent ion current, which is detected by a secondary electron multiplier. With infrared transmissible windows on both sides, the membrane was heated by a focused halogen radiation heater and the temperature as well as the temperature distribution was recorded by a heat sensitive camera. The membranes, illustrated in Figure 1b, were water jet cut (Ø = 30 mm) from a 0.2 mm thick steel foil (AISI 430) and coated as described before. The uncoated back sides were corundum blasted to increase the absorption of infrared radiation. The membrane was mounted with conical copper gaskets with the coating facing the high pressure side. After a minimum base pressure of 1 × 10^−5^ Pa was reached, the measurement was started. 

To investigate the hydrogen barrier properties, the isotope deuterium was employed in order to avoid interpretation ambiguities due to contaminations with residual gases or water molecules. The permeation measurements were performed close to thermodynamic equilibrium. First, deuterium was injected on the atmospheric pressure side. Then the membrane temperature was set to a maximum and was reduced stepwise when a constant ion current was reached. The ion current of the atomic mass m(D_2_) = 4 was recorded. The permeation reduction factor was calculated by (2) using the steady state values of the ion currents of a non-coated sample as a reference.

## 3. Results and Discussion

### 3.1. Oxidation 

To investigate the influence of the oxygen exposure at high temperatures on the phase composition, XRD diffractograms were recorded for different exposure times and compared to the pristine sample. In Figure 2a, the diffractogram of the as-synthesized coating reveals an almost phase pure Ti_2_AlN coating having a strong {0001}-texture. The peak at 42.5° is attributed to the (200) lattice plane of TiN. The Fe-bcc peaks at 44.6° and 65.0° are assigned to the steel substrate. Diffractograms of the oxidized samples are depicted in Figure 2 with an offset for better visibility. 

These phase compositions appear almost unaltered upon thermal exposure. Only in the 2Θ-region between 42°–43° a slight change in the peak position is visible. This region is depicted in detail in Figure 2b. Due to broad peak widths and weak angle dependent interferences, the signals from TiN and α-Al_2_O_3_ cannot be clearly distinguished. Further ambiguities arise due to the small TGO layer thickness and its possibly amorphous structure. Hence, further surface sensitive Raman analysis was performed (see Figure 3). The Raman spectra of the coatings in Figure 3a still feature the characteristic Raman peaks for Ti_2_AlN despite the oxidized surface, though an increase in the background is detected. The broad background and the peak between 500 cm^−1^– 600 cm^−1^ might stem from the formation of surface oxides and/or oxycarbides [21] as well as from the formation of TiN close to the surface due to Al depletion [22]. Titanium oxides like anatase and rutile, which were reported in [23] after the oxidation of Ti_2_AlN coatings at 750 °C, are not detected. 

The fluorescence spectra in Figure 3b exhibit an increase in the background with an increasing oxidation time. Particularly after 100 h of oxidation, the formation of distinct peaks close to 1379 cm^−^^1^ and 1402 cm^−^^1^ can be detected. According to literature, these peaks are attributed to the fluorescence of Cr^3+^ and Fe^3+^ impurities in an α-Al_2_O_3_ environment [24,25,26].

After oxidation, XPS depth profiles of all coatings were created by convoluting the distribution of the binding energies. The underlying XPS spectra are not shown herein. In Figure 4 profiles of a sample oxidized for 100 h (a) and of an uncoated substrate (b) are presented for comparison. In the case of the MAX phase coating Figure 4a, oxidic Al2p bonds with a maximum in the energy of 74.3 eV were detected and ascribed to the formation of Al_2_O_3_ at the sample surface. To a smaller extend of about 8 at%, oxidic Ti2p bonds with a maximum in the energy of 458.2 eV were detected, which were distributed over two regional maxima located in a depth of ~45 nm in the Al_2_O_3_ scale and at the interface of Ti_2_AlN/TGO. At the interface, the shift of the Ti2p-signal to nitridic binding energies of 454.3 eV and the shift of the Al2p-signal to metallic binding energies of 72.3 eV revealed the transition to the Ti_2_AlN phase. With only ~16 at% of Al2p bonds and sustaining increasing signals at a depth of 200 nm, a transition regime close to the interface is observed where an Al depletion exists. The thickness d_Ox_ of the TGO scale was determined by the decrease of the O1s signal, and set to the sputter depth where the O-signal fell below 50% of the original ratio. 

The numerical fit of the values for d_Ox_ according to (1) is plotted in Figure 5. The errors of the oxide thicknesses were estimated to 10 nm resulting from a temporal variation in the sputter rate during XPS measurements. The growth factor of the TGO was calculated to $k_{n}=402\pm 17\;\frac{\mathrm{mol}}{m\times s}$ with a quality factor of $R^{2}=0.9794$. The quality of the fit argues for the suitability of the mathematical description by (1 for the oxidation kinetics. However, no conclusion can be drawn so far as to whether the O or the Al diffuses along the grain boundaries to the oxidizing interface according to the above described model of G.M. Song et al.

The spectra of an uncoated ferritic steel substrate after 100h at 700 °C in Figure 4b revealed the formation of a TGO consisting of of Cr-, Mn- and Fe-oxides. The thickness of the TGO was calculated to 540 nm, which compares to approximately 100 nm in the case of a coated substrate.

The EPMA images of the sample oxidized for 100 h at 700 °C depicted in Figure 6 represent the elemental distributions of Ti (b), Al (c), N (d), O (e), Ni (f) and Fe (g), where the colors indicate the normalized elemental concentration. The measured distribution of Ti, Al and N across the coating thickness features a depletion in Ti and Al at the interface of Ti_2_AlN/TGO and in the subsurface region. Accordingly, the substrate is locally enriched by Al and N and the formation of precipitates perpendicular to the surface is visible. In such areas, only a minor Fe-concentration is measured, as the microprobe signal is always to normalized 100% for all elements. The inward diffusion of Al and N is accompanying the outward diffusion of Fe into the coating according to the Fe elemental distribution map. Besides the thin oxide scale, which formed on top of the MAX phase coating, oxygen can be detected within the Ni-plating. This is caused by the formation of a longitudinal crack within the Ni-plating during the preparation of the cross-section.

The strong interdiffusion of the weakly bound A-element of MAX phases with the substrate is known to be a crucial aspect, when it comes to the chemical stability in high temperature applications [23]. Therefore, the interdiffusion should be suppressed by applying additional barrier films against Al diffusion between the substrate and coating. Further loss of the A-element also occurs during oxidation and annealing in vacuum due to Al_2_O_3_ formation and evaporation. In [27], Zhang et al. calculated that the Ti_2_AlN MAX phase lattice structure is capable of accommodating Al vacancies down to a Ti_2_Al_0.75_N stoichiometry.

### 3.2. Hydrogen Permeation

The D_2_ ion current was measured by secondary ion mass spectrometry using the setup illustrated in Figure 1. The influence of the coatings on the permeation was investigated in a state close to the thermodynamic equilibrium. Three different membranes were investigated: the uncoated substrate material (substrate), the substrate coated with 2.7 µm of Ti_2_AlN (substrate+Ti_2_AlN) and the substrate coated with 2.7 µm of Ti_2_AlN with a subsequent oxidation for 20 h at 700 °C (substrate+Ti_2_AlN+TGO). In Figure 7, the D_2_ ion currents are plotted in the temperature range from about 50 °C to 400 °C. Three measuring cycles were performed for each membrane. The results of the different cycles are denoted by the different symbols in the graph (square, circle and triangle). The small variance in the data points confirms the reproducibility of the permeation induced ion current.

The diffusion through all three membranes follows an Arrhenius type behavior, which confirms the assumption of a diffusion-controlled permeation. The deposition of a 2.7 µm thick Ti_2_AlN coating already reduces the permeation of deuterium. The PRF at 300 °C is calculated to a factor of 45. As optical investigations on the coating after the measurements still revealed some minor cracks in the coating, the PRF might still be lower for a defect-free MAX-phase coating. However, the formation of the oxide scale leads to a further, significant reduction of the permeation. With an oxidation of the coated steel membrane for 20 h at 700 °C, the formation of TGO of 80 nm thickness is expected, compare Figure 5. With respect to the uncoated steel membrane, a PRF of about 3700 was achieved. This reduction can be explained by the low solubility of hydrogen in the α-Al_2_O_3_ phase as well as by the potential healing of small defects in Ti_2_AlN, which blocks alternative migration paths with low energy barriers. The reduction of three orders of magnitude strongly supports the initially assumed suitability of Ti_2_AlN coatings as high temperature hydrogen diffusion barriers.

## 4. Conclusions and Outlook

Ti_2_AlN coatings were synthesized on ferritic steel samples by a repeated deposition of Ti/AlN double layers and a subsequent annealing in vacuum. The oxidation experiments at 700 °C in air revealed the formation of a thin TGO at the sample surface consisting mainly of α-Al_2_O_3_. By analyzing the thickness of the TGO, the kinetic confirms the findings of G.M. Song et al., which are described by a growth in thickness with a time dependency of ~t^1/4^. Thereby, a thin protective oxide is quickly formed after exposure to air but further growth is strongly hindered by a slow diffusion of migrating particles through the dense oxide. It was shown that the TGO not only serves as an effective protective layer against further oxidation, but also serves as a diffusion barrier against hydrogen. Whereas a 2.7 µm thin Ti_2_AlN coating reduces the permeation of deuterium to a factor of 45, the formation of an α-Al_2_O_3_ scale further reduces the permeation within three orders of magnitude. The healing of coating defects like pores and cracks at elevated temperatures upon oxidation is seen as an additional advantage of thermally grown diffusion barriers in comparison to directly deposited barrier coatings.

Further investigations need to focus on the interdiffusion process at the interface of coating and substrate in order to reduce the loss of the Al, which is required for the formation of α-Al_2_O_3._ Finding an optimum thickness of the TGO, which significantly reduces the hydrogen permeation and at the same time exhibits a sufficient thermal and mechanical stability is the crucial task for the utilization of Ti_2_AlN as protective coatings in industrial applications. 

In summary, Al containing MAX phase coatings, which tend to form a dense α-Al_2_O_3_ on the surface upon oxidation, seems to be effective protective coatings in high temperature applications, where oxygen and hydrogen corrode the substrate material.

## Figures and Tables

**Figure 1 materials-13-02085-f001:**
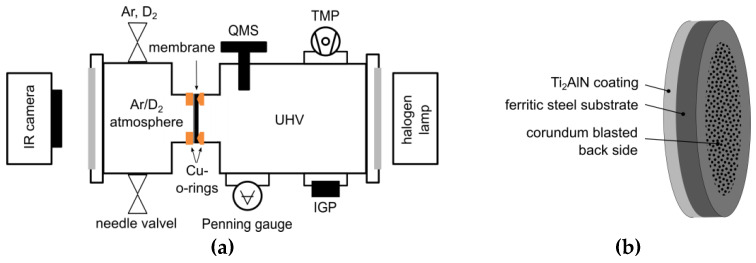
(**a**) Schematic illustration of the hydrogen permeation test rig with quadrupole mass spectrometer (QMS), turbomolecular pump (TMP) and ion getter pump (IGP). (**b**) Schematic illustration of the coated ferritic steel membrane.

**Figure 2 materials-13-02085-f002:**
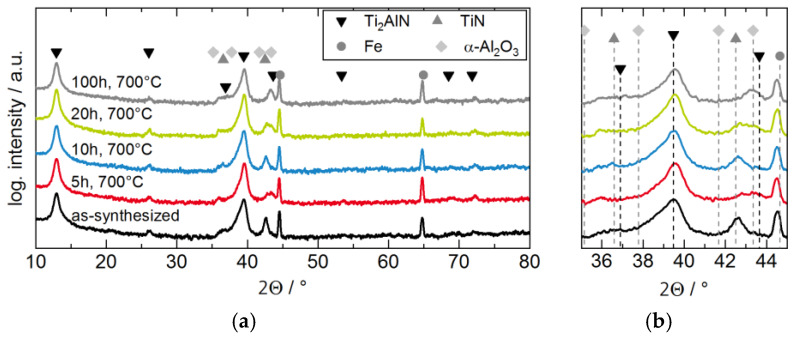
XRD diffractograms of as-synthesized and oxidized Ti_2_AlN coatings on ferritic steel samples: (**a**) overview and (**b**) enlarged region around 2θ° ≈ 40°.

**Figure 3 materials-13-02085-f003:**
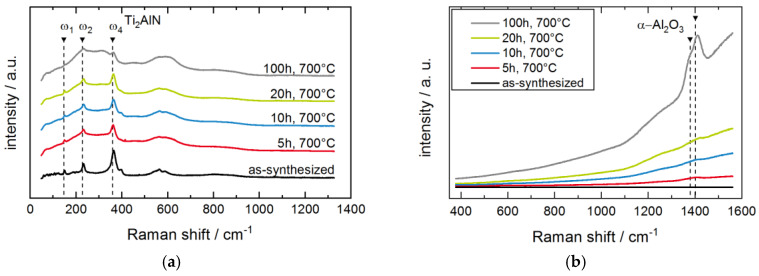
(**a**) Raman spectra (λ_Nd:YAG_ = 532 nm) of the Ti_2_AlN coatings before and after oxidation. (**b**) Raman fluorescence spectra (λ_HeNec_ = 633 nm) of the Ti_2_AlN coatings before and after oxidation.

**Figure 4 materials-13-02085-f004:**
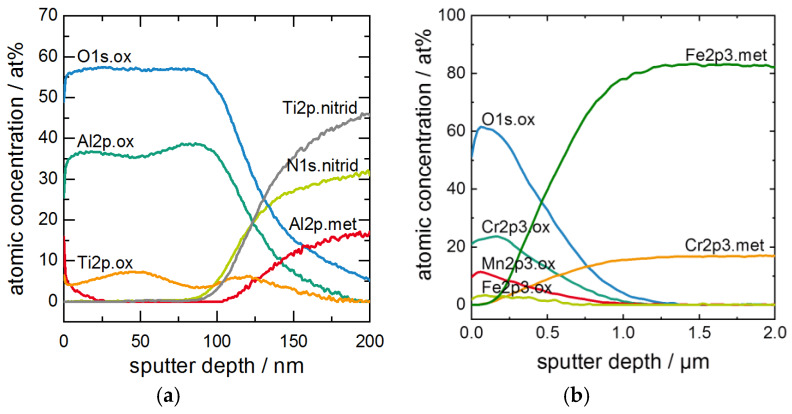
XPS depth profiles after oxidation for 100 h at 700 °C of (**a**) Ti_2_AlN coating, and (**b**) uncoated ferritic steel substrate.

**Figure 5 materials-13-02085-f005:**
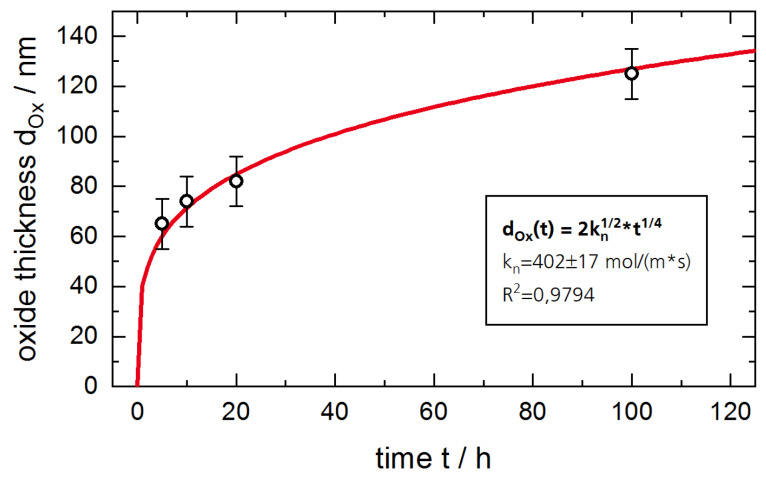
Measured oxide thicknesses by XPS depth profiles of the Ti_2_AlN coatings after oxidation. Underlying fit was performed using (1).

**Figure 6 materials-13-02085-f006:**
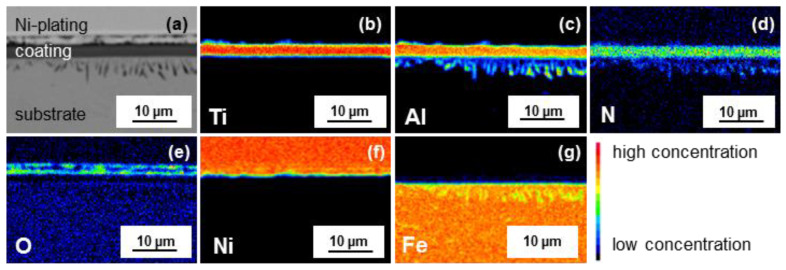
Scanning electron microscope image (**a**) and electron probe micro analysis (EPMA) scans of the cross-section after oxidation for 100 h at 700 °C for Ti (**b**), Al (**c**), N (**d**), O (**e**), Ni (**f**) and Fe (**g**).

**Figure 7 materials-13-02085-f007:**
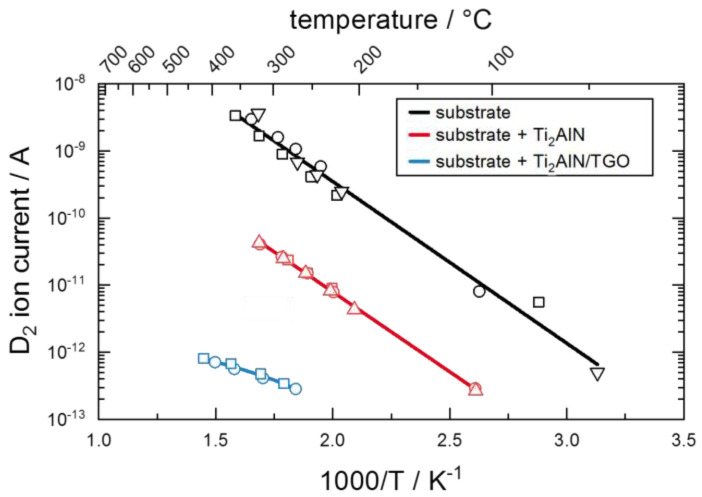
Arrhenius Plot of D_2_ ion currents of ferritic membranes with and without Ti_2_AlN or Ti_2_AlN+thermal grown oxide (TGO) coating. The different symbols on each line refer to results of consecutive measuring cycles. The quasi-linear fit is performed to illustrate the Arrhenius type behavior.

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
