# Peer review of "Investigations of the Deuterium Permeability of As-Deposited and Oxidized Ti2AlN Coatings"

_materials, 2020, doi:10.3390/ma13092085_

Round 1

Reviewer 1 Report

First, allow me to congratulate you on your work. Second please direct your attention to the following small editing errors within the text:

  • Line 90 : Oxidation procedure (not procedure) and analysis ;
  • Line 125: were water jet cut (not cutted);
  • Line 177: in the version of the document I received line 177 contains figure 5 which also appears at line 187;
  • Line 189: contains error message for reference not found, please address this issue.

Reviewer 2 Report

I think the basic idea of the paper is interesting and sound - the improvement of surface conditioning of materials to prevent of  hydrogen diffusion induced embrittlement. Ti2AlN coating was selected and tested. The coatings were oxidized at 700 °C for 5 h, 10 h, 20 h and 100 h in a muffle furnace. The structure and phase analysis was performed by XRD and Ramen spectroscopy To assess the depth profiles the XPS was applied.

Due to some inaccuracies I suggest major revision.

Some detailed comments one can find below:

  • As I remember now we have 18 groups of elements written in Arabic numerals, not in Roman ones.
  • The abbreviation DFT should be explained.
  • SI units should be applied - page 3, line 117,129.
  • Page 6, lines between 176-177 the Figure 5 is located and the second time on the same page but between the lines 186-187.
  • Figure 4 look rather like GDOES spectrum not XPS. It would be interesting to see typical XPS spectrum of Al, Ti and O elements to see the metal-oxide bonding.
  • In caption of Figure 5 is comment in German, it should be changed.
  • 7 - three symbols are shown: circle, square and triangle. What do they mean?
  • There is a repetition in the sentence "The PRF at 300 °C is calculated to PRF = 45."
  • Authors should write that calculated kinetic of TGO confirms findings of G.M. Song. Presented here form, as " The oxide layer thickness increased with a time dependency of ~t1/4." suggests that it is Authors findings.
  • The References should be standarized.

Reviewer 3 Report

The presented paper concerns the Investigations of the deuterium permeability of as-deposited and oxidized Ti2AlN coatings. The paper for me is interesting but before publishing in the journal ,,Materials" the following issues should be corrected:

  1. The Autor should add microstructure and chemical composition of AISI 430 ferritic stainless steel in chapter -experimental details.
  2. Lack of information in the manuscript, what is thermal stability of AlN, the Autor szhould add information in the text
  3. What purity targets were used to apply the Ti and AlN coatings?
  4. How many Ti and AlN coatings were alternately applied to the steel substrate
  5. Which first coating was applied to the steel substrate Ti or AlN. In this experiment is very important
  6. It is not describe peak between 500-600 in figure 3a, why?
  7. The Autor doesn't observe TiO2, after oxidation process, for me the analysis is not complite. What is the enthalpy of TiO, TiO2 and TiN formation?
  8. For me it is not possible to obtain thermochemical stability of a-Al2O3 bellow temperature 1000C. Probably the a-al2O3 contains Cr. The Autor should explain this result
  9. Measured oxide thicknesses by XPS depth profiles of the Ti2AlN coatings after oxidation. Underlying 188 fit was performed using Fehler! Verweisquelle konnte nicht gefunden werden..what is?

Reviewer 4 Report

Here is the list of my comments:

-I consider that the conclusion should be deeply explained and in more clear form.

- it is a recommendation to add in chapter 3.1. the chemical composition of the coating and steel substrate. Thus, it Will be Good to explain the conclusions more clearly and schematically.

Round 2

Reviewer 2 Report

All my comments have been taken into account.
I just have a comment about the uniformity of References. Some of the Journals are shown in the full name and some in the abbreviation, [6,8,12,23, ...]. In Ref.[10] please delete "In print" and provide full bibliographic data.

After correcting the manuscript can be accepted for publication